# Cloning and Characterization of a Thermostable Endolysin of Bacteriophage TP-84 as a Potential Disinfectant and Biofilm-Removing Biological Agent

**DOI:** 10.3390/ijms23147612

**Published:** 2022-07-09

**Authors:** Joanna Żebrowska, Olga Żołnierkiewicz, Małgorzata Ponikowska, Michał Puchalski, Natalia Krawczun, Joanna Makowska, Piotr Skowron

**Affiliations:** 1Department of Molecular Biotechnology, Faculty of Chemistry, University of Gdansk, 80-309 Gdansk, Poland; olga.zolnierkiewicz@ug.edu.pl (O.Ż.); malgorzata.ponikowska@ug.edu.pl (M.P.); natalia.krawczun@ug.edu.pl (N.K.); piotr.skowron@ug.edu.pl (P.S.); 2Laboratory of Biopolymers Structure, Intercollegiate Faculty of Biotechnology UG&MUG, University of Gdansk, 80-309 Gdansk, Poland; michal.puchalski@ug.edu.pl; 3Department of General and Inorganic Chemistry, Faculty of Chemistry, University of Gdansk, 80-309 Gdansk, Poland; joanna.makowska@ug.edu.pl

**Keywords:** bacteriophage, TP-84, bacteria lysis, thermostable endolysin, lysins, peptidoglycan, cell wall, biofilm, disinfection, *Geobacillus*

## Abstract

The obligatory step in the life cycle of a lytic bacteriophage is the release of its progeny particles from infected bacterial cells. The main barrier to overcome is the cell wall, composed of crosslinked peptidoglycan, which counteracts the pressure prevailing in the cytoplasm and protects the cell against osmotic lysis and mechanical damage. Bacteriophages have developed two strategies leading to the release of progeny particles: the inhibition of peptidoglycan synthesis and enzymatic cleavage by a bacteriophage-coded endolysin. In this study, we cloned and investigated the TP84_28 endolysin of the bacteriophage TP-84, which infects thermophilic *Geobacillus stearothermophilus*, determined the enzymatic characteristics, and initially evaluated the endolysin application as a non-invasive agent for disinfecting surfaces, including those exposed to high temperatures. Both the native and recombinant TP84_28 endolysins, obtained through the *Escherichia coli* T7-lac expression system, are highly thermostable and retain trace activity after incubation at 100 °C for 30 min. The proteins exhibit strong bacterial wall digestion activity up to 77.6 °C, decreasing to marginal activity at ambient temperatures. We assayed the lysis of various types of bacteria using TP84_28 endolysins: Gram-positive, Gram-negative, encapsulated, and pathogenic. Significant lytic activity was observed on the thermophilic and mesophilic Gram-positive bacteria and, to a lesser extent, on the thermophilic and mesophilic Gram-negative bacteria. The thermostable TP84_28 endolysin seems to be a promising mild agent for disinfecting surfaces exposed to high temperatures.

## 1. Introduction

Despite the fact that thermophilic living conditions seem to be unfriendly for most living creatures, these ecosystems are inhabited by numerous microorganisms. Moreover, those thermophiles mirror the relationships that exist between mesophilic species, especially those between bacteriophages and bacteria. Such thermophilic biotic systems can be isolated from various thermophilic sources, in which the ambient temperature rises as a result of both natural processes and human activity (hot springs and the surrounding soil, hydrothermal vents, soil near volcanoes, compost heaps, greenhouse soil, power-plant cooling units, wastewater, and others) [1].

The bacteriophage TP-84 was isolated from greenhouse soil in 1952, becoming the *Siphoviridae* family and a member of the genus *Tp84virus* [2]. It contains an icosahedral capsid and a long, flexible tail. The host of the bacteriophage TP-84 is the thermophilic Gram-positive *Bacillus stearothermophilus* (currently *Geobacillus stearothermophilus*) strain 2184. TP-84 only infects several *G. stearothermophilus* strains [2], which reach a length of 2 to 3.5 µm and diameter of 0.6 to 1.0 µm; they contain a thick capsule and are usually found as single cells. 

The lytic cycle of TP-84 lasts 22–24 min under optimal conditions: (i) propagation at a 55–58 °C temperature, (ii) media supplemented with glucose/fructose, and (iii) the presence of Ca^2+^ ions [2,3]. Remarkably, it can grow through an enormous temperature range of 30–80 °C [4]. This would overlap with the temperature ranges in the environmental niche of *G. stearothermophilus* strains, the majority of which grow at 40–70 °C and a pH of 6–8 [5]. 

Virulent TP-84 follows a general growth pattern: (i) recognition and specific adsorption to their host, (ii) introduction of their genome into the cell, (iii) replication, (iv) transcription, (v) coordinated biosynthesis of bacteriophage proteins, (vi) maturation, (vii) assembly of virions, and (viii) lysis of the bacterial cell and release of progeny virions [6]. In order for replicated bacteriophages to be released for the continued infection of the bacterial population, external barriers must be overcome, where each layer is targeted by a different enzyme (Table 1). 

The greatest obstacle for the new lytic TP-84 to overcome is the peptidoglycan layer (containing about 40 layers of peptidoglycan) [11,21], a stable structure that counteracts the pressure prevailing in the cytoplasm and protects the cell from osmotic lysis and mechanical damage [7,8]. Lytic bacteriophages have developed two general strategies leading to the release of young units grown by destroying the peptidoglycan barrier: the inhibition of its synthesis or damage from enzymatic cleavage, mainly by the action of endolysins [8,9,10]. 

There is also an alternative lytic system, based on endolysins, with a signal–arrest–release (SAR) sequence at the N-terminal ends of certain bacteriophage endolysins, allowing them to pass through the cell membrane without the holins by utilizing the bacterial secretory system. The SAR domain then remains anchored in the membrane while the catalytic part of the enzyme is already exposed on the outer membrane surface in an inactive form until it folds into a catalytically active state and is released into the periplasm of Gram-negative bacteria [7,8,10]. 

Gram-positive bacteria endolysins have evolved into the modular structure in which catalytic activity and substrate recognition are divided into separate functional domains, referred to as cell wall-binding domains (CBD) and enzymatically active domains. The CBD binds the peptidoglycan and remains associated with cell wall debris following cell lysis, which likely prevents the diffusion and subsequent damage of the surrounding intact cells that have not yet been infected by the bacteriophage [10,15]. In turn, Gram-negative bacteria are resistant to such damage by utilizing an external membrane barrier. This is probably why endolysins from Gram-negative host-specific bacteriophages are usually small globular proteins with a single domain of 15–20 kDa molecular weight, without a CBD domain [12,22]. 

The use of endolysins as potential antimicrobial candidates is an interesting preventive and/or therapeutic alternative or supplement to conventional antibiotic therapy [11,23]. In this work, we have cloned and characterized thermostable bacteriophage TP-84 TP-84_28 endolysin and conducted an initial evaluation of its potential applications as a microbial population-controlling agent.

Endolysins are proteins with strong antibacterial properties. A very small amount of endolysin is able to eliminate bacteria from a bacterial suspension or bacterial biofilm. Of all the benefits endolysins offer, the most compelling aspect of these enzymes is their ability to fight antibiotic-resistant bacteria. The effectiveness of thermostatic endolysins has been demonstrated against strains of bacteria resistant to many drugs, such as penicillin and vancomycin. High temperatures deactivate most of the available antibiotics, a fact that may make endolysin cocktails particularly useful against thermophilic bacteria [24]. Bacteriophage endolysins are a promising antimicrobial weapon because they show strong and rapid bactericidal and anti-biofilm activity, low induced resistance and cellular toxicity, and synergy with common antibiotics. Remarkably, their narrow spectrum of antimicrobial activity ensures precision in killing the target bacteria without disturbing the beneficial microflora. In addition, advances in membrane-penetrating nanomaterial technology will provide a better endolysin delivery strategy. It is a promising approach to antimicrobial therapy in the future [25].

Endolysins exhibit a rather narrow host range; therefore, applications of the TP84_28 endolysin would likely focus on areas where bacteria related to the original TP-84 host are a problem. The food industry is definitely one such area, as numerous *Bacillus* species have been shown to cause food spoilage. Many endolysins that are active against bacteria that cause foodborne infections, such as *S. aureus*, *Listeria monocytogenes*, or *Clostridium perfrigens*, have already been tested as food additives and have demonstrated very promising results [26]. However, the food industry must protect against not only bacteria that are directly pathogenic to humans, but also those responsible for spoilage.

Evidence for the safety of endolysins in humans is sparse thus far, but a few completed studies have not raised any concerns [26]. Regarding the effects on humans, it should be noted that depending on the food-processing stage at which endolysin is applied, it may simply be destroyed after fulfilling its mission (i.e., by temperature or by being digested by an enzyme in the gastrointestinal tract), and therefore will not directly interact with humans. Since thermostable TP-84 endolysin undergoes deactivation at very high temperatures and is not self-replicating, it should not pose a problem as a food-processing agent. On the other hand, if the endolysin is added as a food preservative after processing, any such use should be preceded by an extensive human safety study. Thermostable endolysins are heat-resistant and active over a wide pH range, which makes them a very attractive preservative for all types of food, especially those that do not require refrigeration. However, this raises the question of how the long-term stability of thermostable endolysins would look if they were used as a canned food additive.

Examples of food categories where endolysin can be useful are listed below. 

Bakery products—Various species of *Bacillus* (including *B. pumilus*, *B. amyloliquefaciens*, and *B. licheniformis*) are responsible for spoilage in bakery products. These bacteria are usually present in the raw materials for bakery products and are able to spore under unfavorable conditions [27]. The spores can withstand high temperatures, so processing may not always eliminate contamination. Therefore, if an enzyme that actively digests the bacteria is added to the bakery product during processing, spore formation and the premature deterioration of the final product can be prevented [28].

Milk products—*B. cereus* is considered the species of bacteria that is most often responsible for the spoilage of dairy products. Not only does it destroy the properties of dairy products, but it can cause food poisoning with symptoms such as vomiting and diarrhea. *B. subtilis*, *B. amuloliquefaciens*, *B. smithii*, *Geo-bacillus pallidus*, and *G. sterothermophilus* are other species that are typically responsible for the spoilage of dairy products [27]. Thermostable endolysin can therefore potentially be used to eliminate these bacteria. Activity against *B. subtilis*, *B cereus*, and *G. sterothermophilus* has already been confirmed in this study; thus, it can be expected that TP-84 endolysin will be active against any other related species, though further tests should be performed to prove its antimicrobial activity.

UHT milk—This is a product that is processed at an ultra-high temperature (130 °C) for 4 s. This treatment eliminates most of the potential spoilage species, though *G. stearothermophilus* is able to survive as spores [27]. Pre-treating milk with thermostable endolysin can help to reduce the number of bacteria capable of forming spores when forced to do so by unfavorable conditions. Adding purified endolysin after sterilization can also be considered.

Dehydrated milk—*G. stearothermophilus* is also the main species found in powdered milk [27]. As mentioned above, pretreating the milk before pulverizing it can help reduce spores and bacterial contamination when the milk is reconstituted into liquid form.

Low-acid canned food—*G. stearothermophilus* is one of the leading causes of spoilage in low-acid canned food [27]. The addition of a thermostable endolysin active against this species could prevent food spoilage, but more data would be needed on both human safety and the long-term stability of the endolysin.

## 2. Results

### 2.1. Bioinformatic Analysis of TP84_28 Endolysin Gene

Bioinformatic analysis of the bacteriophage TP-84′s genome showed a clear division into three functional bacteriophage gene groups/clusters. The first cluster of genes is associated with the assembly of TP-84 capsids. The second gene cluster is responsible for nucleotide metabolism and DNA processing. The third cluster includes gene coding for the protein-degrading envelope, cell wall, and cytoplasmic membrane—TP84_26 glycosylase-depolymerase, TP84_27 holin, and TP84_28 endolysin—thus allowing for the release of progeny bacteriophage particles [2]. Most bacteriophage endolysins can be produced as recombinant variants in bacterial cells because their aa sequences do not contain a transport signal for passage across a cytoplasmic membrane. In the case of endolysins with an SAR transport signal at the N-terminus [7], biosynthesis of their recombinant variants in bacterial cells is problematic. There is no information about a preserved consensus sequence for SAR regions; it is only known that these are hydrophobic regions with a sequence that may resemble transmembrane domains. The TP84_28 gene is preceded by the holin gene sequence, which may functionally replace SAR action [2]. On the other hand, the fact that the TP-84_28 endolysin has a relatively large molecular size suggests that the enzyme may have more capabilities than digesting peptidoglycan. Bioinformatic analysis showed the existence of highly conserved domains. The largest motifs indicate the presence of domains that are located in lysins or proteins containing the GH25 domain. TP84_28 endolysin’s largest domain is GH25_Lyc-like, located within 21–206 aa (Appendix A). Fragments of 43 aa in the C-terminus are designated as LysM superfamily domains, involved in peptidoglycan binding in bacteria and chitin in eukaryotes. These domains degrade the bacterial cell wall, but the proteins involved in many other biological functions also contain this domain [29,30].

### 2.2. Biosynthesis and Isolation of Native and Recombinant TP84_28 Endolysins

#### 2.2.1. Cloning, Overexpression, and Purification of Recombinant TP84_28 Endolysin

The PCR-amplified TP84_28 endolysin-coding sequence was cloned into the modified pET21d(+) vector (pET21dHis) in perfect fusion with the vector’s START codon. The pET21d_His vector was designed to introduce His6-tag at the N-terminus of a recombinant protein along with the immunodetection enhancer MRGS. The coding DNA sequence was introduced upstream of the NcoI site, originally present in the pET21d(+) vector. Cloning of the TP-84_28 endolysin gene resulted in the fusion protein, containing an MMRGSHHHHHH segment at the recombinant enzyme N-terminus, introduced by the vector (Appendix A). The recombinant clones pET21dHis_TP84_28 were confirmed by sequencing 7 “positive” recombinant plasmids that were expressed in three different *E. coli* strains: BL21(DE3), BL21(DE3) pLysS, and ARCTICA. Interestingly, only the strain BL21(DE3) expressed TP84_28 endolysin. Furthermore, no decreases in bacterial culture optical density (OD) after IPTG induction were observed in the other strains (Figure 1a), which would be expected in the expression of such a highly toxic protein. The pET21dHis_TP84_28 clones in *E. coli* BL21(DE3) also showed a characteristic rim around the colonies of the transformants (Figure 1b). 

A simple purification procedure for recombinant TP84_28 endolysin was devised, containing PEI and ammonium sulfate-selective precipitations followed by metal affinity purification. The protein was eluted with 450–500 mM imidazole. The final purification was obtained by ion exchange chromatography on DEAE–Sepharose with 200–300 mM NaCl elution. A purity of 98% was apparent in the SDS-PAGE of the recombinant protein (Figure 2a). The purified preparation can be stored at 4 °C for more than 2 months and at −80 °C with added glycerol (10%) or at −20 °C (50% glycerol) for over 6 months. The presence of the correct recombinant protein was confirmed by Western blotting analysis using anti-His6 antibodies (Figure 2b). The enzymatic activity of the recombinant TP84_28 endolysin was convincingly confirmed in intact *G. stearothermophilus* strain 10 cells using zymographic assay (Figure 2c) and a diffusion test (data not shown). Zymographic assay clearly revealed the complete lysis of bacterial cells immobilized in the polyacrylamide at the location corresponding to the position of recombinant TP84_28 in SDS-PAGE and Western blotting (Figure 2). From 1 L of the expression culture (1 g cells), we obtained 1.24 mg enzymatically active recombinant protein at the concentration of 0.2 µg/µL (4.44 µM).

#### 2.2.2. Isolation and Purification of Native TP84_28 Endolysin

The native TP84_28 was purified from bacterial cell lysate after bacteriophage TP-84 infection. The native TP84_28 endolysin, present in the cleared supernatant, was purified using cation exchange CM-Sepharose, eluted with 500 mM NaCl, and further passed through strong anion-exchange Q-Sepharose, which acts as a “negative” purification stage. The preparation was filtered on a membrane with a 100-kDa cut-off and then concentrated on a Viva spin filter with a 10-kDa cut-off. From 1 L of the lysate, 10 µg native TP84_28 endolysin was obtained (Figure 2a). Its activity was confirmed by a diffusion test and zymografic analysis and was then compared to the recombinant variant (Figure 2c). The MS/MS-LC tryptic peptide sequences for the purified enzyme were in agreement with the bioinformatically predicted aa sequence of TP84_28 endolysin, as based on the genomic ORF translation (Appendix A). The native enzyme is produced in small amounts during the TP-84 life cycle—from 2 L of lysate, we obtained 0.15 mg protein, at a concentration of 0.0195 µg/µL (0.44 µM). Obtaining large amounts of native TP84_28 endolysin would require a scaled-up and modified procedure.

### 2.3. Properties of TP84_28 Endolysin

#### 2.3.1. Activity of Recombinant versus Native TP84_28 Endolysin

A spot (diffusion) test and zymographic test confirmed the muralytic activity of both proteins. The enzymes were spotted onto top agar with an overnight culture of *G. stearothermophilus* strain 10. This resulted in efficient lysis, leaving transparent spots on bacterial lawns. The enzymatic activity of those muralytic enzymes was assayed semi-quantitatively using a zymogram containing incorporated *G. stearothermophilus* host cells. Denaturing zymogram analysis with initial separation on SDS-PAGE of both endolysin preparations showed gel-embedded cell lysis associated with bands of the individual proteins, which was observed as clear zones/bands within a gel after SDS removal and renaturation. To confirm the location of clear zones on the zymogram, corresponding to the TP84_28 endolysin band position, an identical SDS-PAGE process—apart from the added bacteria—was run and stained with Coomassie blue. The location of a protein band, exhibiting lytic activity on the zymograms, precisely correlated with the position of a Coomassie blue-stained band in SDS-PAGE and was in agreement with the molecular-weight calculations based on the positions of the protein marker bands. 

Interestingly, the plain removal of SDS by washing the gel in distilled water for 2 h at room temperature without any additional buffer components was sufficient to induce TP84_28 endolysin refolding into an enzymatically active state (Figure 2c). Furthermore, alternative methods were employed to assess the enzyme’s activity in its native state without denaturation/renaturation in order to evaluate the proper folding of thermostable TP84_28 endolysin, expressed in mesophilic *E. coli*. The activities of the recombinant and native enzymes were compared by measuring the decrease in OD after adding the purified protein preparations to the TRA reaction, conducted with *G. stearothermophilus* cells at 55 °C. Both enzymes caused the rapid degradation of the cells, which was evident in the first 10 min. However, after 10 min of incubation, both enzymes were inhibited because between 10 and 30 min of the process, no further significant decrease in activity was noticed (Figure 3). Due to the minute amounts of native TP84_28 endolysin produced by the infected *G. stearothermophilus* cells, this enzyme preparation was used at a much lower concentration than the recombinant enzyme; this variant caused faster lysis of the cells, as expected (Figure 3). Nevertheless, the combined results from zymographic analysis and the TRA reactions, adjusted for the different enzyme concentrations, point to the conclusion that both variants exhibit very similar specific activity. 

#### 2.3.2. Thermostability of Recombinant TP84_28 Endolysin

Thermostability tests were also based on measurements of the enzyme samples’ relative activities at various temperatures. No significant reduction in muralytic activity was observed after 30 min of pre-incubation of the enzyme in temperatures ranging from 30 °C to over 70 °C, followed by transferring the samples to an optimal 55 °C. A significant decrease in relative activity was observed starting from 74 °C upwards; however, heating the enzymes up to 100 °C did not cause complete activity loss (Figure 4a). Therefore, measurements of the activity were also conducted after treating the recombinant TP84_28 endolysin under autoclaving conditions (121 °C for 20 min), which resulted in a 100% loss of activity. Further determination of thermostability using CD spectra of recombinant TP-84_28 endolysin indicated that there were no significant changes in the secondary structure in the range of 30 °C to 75 °C. Increasing the temperature from 75 °C to 110 °C induced changes in the enzyme’s structure, as registered at the 208 nm and 222 nm bands (Figure 4b). The CD analysis revealed that the enzyme’s structure changed above 75 °C in correlation with its decreasing activity. DSC analysis—used to determine the precise melting point and unfolding temperature of the protein, recorded in the temperature range of 30 °C to 100 °C in an optimal activity buffer—showed TP84_28 endolysin’s melting point to be 77.6 °C (Figure 4c). Furthermore, the graph in Figure 5 shows that the heat denaturation of recombinant TP84_28 endolysin is irreversible and the characteristic shape of the graph at 90 °C indicates the aggregation of the protein molecules. These thermostability assay results, obtained by three independent methods, indicate that the recombinant enzyme expressed in the mesophile retains the characteristics of the native enzyme and the upper activity and stability limit correspond with the highest temperature that can support a *G. stearothermophilus*/TP-84 system [3].

#### 2.3.3. pH Effect on TP84_28 Endolysin Activity

The influence that the pH of the reaction buffer R has on recombinant TP84_28 endolysin activity on *G. stearothermophilus* cells was measured at 55 °C as the substrate cell suspension’s OD decreased every 2 min. The results were recorded as a graph of OD dependence on incubation time (Figure 5). Muralytic activity of recombinant TP84_28 endolysin was observed over a wide pH range, from 4.0 to 10.0, being very low to low in the range of pH 4–6 and increasing to a medium value at pH 6.5. The most intense TP84_28 endolysin activity, expressed as the largest reduction in turbidity, was seen at a pH range of 7.0 to 9.0, with a maximum at 7.5–8.0 and a gradual decrease to a medium value, seen even at pH 10.0.

### 2.4. Lytic Spectrum of TP84_28 Endolysin on Bacterial Substrates 

To examine TP84_28 endolysin substrates and bacterial host specificity, the lytic activity was tested on a variety of natural bacterial substrates: mesophilic, thermophilic, Gram-positive, and Gram-negative. These included *G. stearothermophilus*, *B. stearothermophilus*, *Geobacillus* sp., *B. cereus*, *B. subtilis*, *E. coli*, *Thermus thermophilus*, and *Thermus aquaticus*. Spot assays were conducted by growing test bacteria at their optimal temperatures for up to 24 h until bacterial lawns in apparent logarithmic phase (not overgrown) were well visible. The enzyme samples were then spotted on each lawn and further incubated at 55 °C for 16 h. The muralytic activity was also measured using TRA. Table 2 shows that the lytic activity of TP84_28 endolysin was universal on all of the bacteria tested, though it highly prefers thermophilic Gram-positive bacteria—related to the TP-84 bacteriophage host bacteria *G. stearothermophilus* strain 10, *B. stearothermophilus*, *Geobacillus* ICI—over mesophilic Gram-positive *B. subtilis* and *B. cereus.*

### 2.5. Disruption of G. stearothermophilus Strain 10 and Mesophilic Bacterial Biofilms

The addition of TP84_28 endolysin to a biofilm using *G. stearothermophilus* strain 10 caused a significant inhibition of the formation of bacterial biofilm. Treating the bacteria with 0.2 µg of endolysin reduced biofilm formation by 62.5%, while adding 2 µg of protein reduced biofilm formation by 94.4%, and adding 20 µg TP84_28 resulted in as much as a 99.2% reduction (Figure 6a). The percentage inhibition yield in this experiment was confirmed by CFU/mL analysis comparing the untreated TP84_28 sample to the sample treated with 2 µg protein.

The influence of endolysin TP84_28 on the formation of pathogenic mesophilic bacterial biofilms, which are one of the major problems in medical clinics, was also investigated. Endolysin TP84_28 showed the greatest inhibition of biofilm formation by *P. aureginosa* bacteria (as much as 85.9%); for *S. pyogenes* it was 62.5%, for *E. cloacae* 53.2%, and for *S. aures* 41.7%, but for *Salmonella* sp., no biofilm disruption was observed during 1 h of incubation at 55 °C. It was surprising that the addition of endolysin to *A. baumannii* increased the formation of bacterial biofilm, which may be due to increased bacteria sedimentation, which resulted in a higher OD value (Figure 6b).

## 3. Discussion

Putative endolysin encoded by ORF_28 was identified in the TP-84 genome using BLASTX comparison with sequences deposited in the GenBank database. The signal sequence and transmembrane domain were revealed at the beginning of the putative protein, followed by the catalytic domain; thus, TP84_28 endolysin apparently belongs to the widespread Lys muramidase and 1,4-β-N-acetylmuramidase GH25 family of endolysins. In addition, the analysis of the TP84_28 endolysin using SWISS-Protein model software revealed that 1–245 aa residues comprised the catalytic domain. This portion of the protein had 50.21% sequence identity to autolytic lysozyme and could be a model 4krt.1.A. The remaining portion of the protein (220–394 aa residues) had 30.91% sequence coverage to extracellular protein 6, so a putative model could be 4b8v.1.A. Usually, the endolysins of bacteriophages specific to Gram-positive bacteria have an enzymatically active domain and a CBD domain. Therefore, the second part of the protein was expected to be a CBD of unknown specificity. However, modeling showed that the aa sequence from 233 to 405 was 20.12%, similar to the chitin-binding domain. This suggests that the binding of this domain to bacterial peptidoglycan might be somewhat related to the binding of eukaryotic chitin. If this is the case, and TP84_28 endolysin could have degrading properties for selected eukaryotic cells, it would be quite an intriguing evolutionary connection [29,30]. 

The enzymatic activity of the putative endolysin encoded by ORF TP-84_28 was confirmed by cloning into an *E. coli* pET21d(+)-derived expression vector, purifying the His6-tagged, fusing TP84_28 endolysin, and confirming bacterial cell wall-degrading enzymatic activity. The natural production, and thus purification, yield of native TP84_28 endolysin was very low: only 0.075 mg was obtained from 1 L of TP-84-infected cell lysate. Apparently, this natural biosynthesis is controlled in order to avoid lysis from outside of those host cells that have not yet been infected. However, even in the case of the recombinant enzyme, whose coding ORF was cloned under a very strong T7-lac promoter, the yield of TP84_28 endolysin was unexpectedly low: from 1 L *E. coli* expression culture, only 1.24 mg protein was isolated. Apparently, this is due to the high toxicity of the recombinant TP84_28 endolysin, even though its activity at 37 °C is very low. The protein TP84_28 is fine tuned to hydrolytic activity toward *Geobacillus* bacteria in both spot and zymographic assays. The least activity was observed toward Gram-negative bacteria and hyperthermophilic *Thermus* sp. This is likely a result of the outer membrane in Gram-negative bacteria and lipids or other protective incrustations on hyperthermophiles’ cell walls. Interestingly, TP84_28 endolysin had a greater lytic effect on the *Geobacillus* ICI strain than on TP-84 host bacteria *G. stearothemophilus* strain 10. Nevertheless, the conclusion can be drawn that TP84_28 endolysin is geared toward a specific peptidoglycan structure variant of *Geobacillus*, though with certain reservations, considering that the enzyme naturally acts from the inside of the host cell as opposed to the conditions present during the assays. Overall, TP84_28 endolysin is a very robust enzyme, since the biochemical, CD, and DSC analyses all point to very high thermal stability and activity at elevated temperatures, up to approximately 80 °C. Furthermore, for complete inactivation, autoclaving is needed. The enzyme’s robustness also extends to pH extremes: it maintains its activity over a broad pH range, from 4 to 10. 

These features of TP84_28 endolysin can be of practical importance in the development of a novel type of anti-bacterial biological preparation. The use of bacterial cell wall lytic enzymes as potential antimicrobial candidates is an interesting preventive and/or therapeutic alternative or supplement to conventional antibiotic therapy in the times of increasing bacterial resistance to antibiotics [11,23]. This seems especially promising in the case of Gram-positive bacteria, due to the absence of an outer membrane, thus allowing the unrestricted access of lytic enzymes to a peptidoglycan layer [23]. The potential applications of these enzymes, including thermostable variants, include the control of food-borne pathogens, using purified endolysins as preservatives in food/organic products, decreasing bacterial populations on various surfaces, liquids or aerosols containing endolysin, expression by transgenic plants to prevent bacterial infection, and reducing/eliminating bacterial biofilms that exhibit a high level of tolerance to antibiotics and that pose a problem in human infections, food production, and biotechnology industries [11]. Moreover, TP84_28 endolysin can be used as a surface “sterilizing” agent, even for surfaces that are exposed to high temperatures due to their dark color, direct sun exposure, industrial setting, etc. As mentioned above, TP84_28 is especially attractive for applications in the food industry because it targets the bacteria responsible for the contamination of food products. Therefore, TP84_28 could be also used to clean food-processing plants. Its thermostability makes it a very attractive compound because it can be used in high temperatures. Furthermore, TP84_28 endolysin may be employed as a part of an enzymatic “cocktail” in conjunction with a mesophilic endolysin, such as the very active T4 bacteriophage endolysin, to extend its application range to low temperatures. Such a mixture would be devoid of antibiotic limitations and side effects when applied topically to wounds or infected areas. At present, non-antibiotic antimicrobials represent a major unmet medical need for the treatment of bacterial biofilms. In order to avoid the limitations of antibiotic therapies, a promising new strategy has been proposed based on a new understanding of thermostable bacteriophage endolysins as anti-biofilm agents. Endolysins can potentially destroy bacterial biofilms and kill mixed biofilms, thus providing effective standalone or adjunctive therapies for treating biofilm infections [31].

## 4. Materials and Methods

### 4.1. Bacterial Strains, Plasmid, Media, and Reagents

The bacterial media components were sourced from BTL (Łódź, Poland). *E. coli* TOP10 (F^−^ *mcr*A Δ (*mrr*-*hsd*RMS-*mcr*BC) Φ80*lac*ZΔM15 Δ *lac*X74 *rec*A1 *ara*D139 Δ(*ara^-^leu*)7697 *gal*U *gal*K *rps*L (Str^R^) *end*A1 *nup*G) (Life Technologies, Gaithersburg, MD, USA) was used for *E. coli* plasmid propagation using LB medium (1% tryptose, 0.5% yeast extract, and 1% NaCl). *E. coli* BL21(DE3) (B F^−^ ompT hsdS(r_B_^−^m_B_^−^) dcm^+^ Tet^r^ galλ (DE3) endA Hte), *E. coli* BL21(DE3)pLysS (B F^−^ ompT hsdS(r_B_^−^m_B_^−^) dcm^+^ Tet^r^ galλ (DE3) endA Hte pLysS), and *E. coli* ArcticExpress(DE3) (B F^−^ ompT hsdS(r_B_^−^m_B_^−^) dcm^+^ Tet^r^ galλ (DE3) endA Hte cpn10 cpn60 Gent’) (Agilent Technologies, Palo Alto, CA, USA) were used for gene expression. *Bacillus stearothermophilus*/*G. stearothermophilus* strain 10 host for bacteriophage TP-84 was sourced from the collection of Prof. Piotr Skowron and is identical to the host strain published previously [2,3]. *Geobacillus* ICI was isolated by Prof. Skowron’s team from hot springs in Iceland. The TP-84 host was propagated on TYM medium (2% peptone K, 0.4% yeast extract, 9.8 mM MgCl_2_, 5 mM CaCl_2_, and 0.5% fructose). The biofilm survey and disruption was propagated on TSBg medium (1.7% peptone K, 0.3% peptone SP, 0.5% NaCl, 0.25% K_2_HPO_4_, and 0.25% glucose; pH 7.3) on 6-well, flat-bottom polystyrene microplates (Nest Scientific Biotechnology, Rahway, NJ, USA) and measured on a Victor3 plate reader (PerkinElmer, Waltham, MA, USA). Biofilm-forming bacteria from the Collection of Plasmids and Microorganisms of the Biology Faculty (KPD) included *Enterobacter cloacae* KPD 297 (*E. cloacae*), *Acinetobacter baumannii* CRAB KPD 205 (*A. baumannii*), *Staphylococcus aureus* MRSA KPD 425 (*S. aureus*), *Salmonella* sp. KPD 449 (*Salmonella* sp.), *Pseudomonas aeruginosa* MDR KPD 430 (*P. aeruginosa*), and *Streptococcus pyogenes* KPD 457 (*S. pyogenes*).

The plasmid DNAs were isolated using DNA purification kits from BLIRT (Gdańsk, Poland), A&A Biotechnology (Gdynia, Poland), and Omega bio-tek (Norcross, GA, USA). The NcoI-HF, SalI-HF, and StuI restriction endonucleases were from New England Biolabs (Ipswich, MA, USA) and the Eco31I was from ThermoFisher Scientific (Waltham, MA, USA). The T4 DNA Ligase, dNTPs, 100 bp, 1 kb DNA, protein ladders Pierce^TM^ Unstained Protein MW, and low-melting agarose were from ThermoFisher Scientific Baltics UAB (Vilnus, Lithuania). The Q5^®^ High-Fidelity DNA polymerase and Q5 High GC Enhancer were sourced from New England Biolabs (Ipswich, MA, USA). Marathon DNA polymerase was from A&A Biotechnology (Gdynia, Poland). Agarose was from Bioshop (Burlington, Canada). PCR primers and VivaSpin 100 kDa membrane were from Sigma-Aldrich (St. Louis, MO, USA). Chromatographic media/columns—Ni-NTA Sepharose Fast Flow, DEAE-Sepharose Fast Flow, Superdex 16/60 75 pg, CM-Sepharose, CM-Sephadex C-25, and Q-Sepharose—were from GE Healthcare (Chicago, IL, USA). Horseradish peroxidase, anti-polyHistidine antibodies, and Sigma FAST^TM^ Protease Inhibitor were from Sigma-Aldrich (St. Louis, MO, USA). The DNA sequencing was performed at Eurofins Genomics (Ebersberg, Germany). SnapGene software version 4.1 (http://www.snapgene.com) was used for genetic maps of the DNA vectors and preparing the recombinant constructs. SWISS-Protein model software (https://swissmodel.expasy.org, accessed on April 2022) was used for comparing protein homologies. Other reagents were from Avantor Performance Materials Poland S.A. (Gliwice, Poland), AppliChem Inc. (St. Louis, MO, USA), or Fluka Chemie GmbH (Buchs, Switzerland).

### 4.2. Recombinant TP84_28 Endolysin Production

#### 4.2.1. Cloning of TP-84 Endolysin Gene into pET21d_His Vector

The TP84_28 endolysin gene was amplified using PCR reaction employing Q5^®^ High-Fidelity DNA polymerase, TP-84 lysate as a DNA template, and the following primers: forward 5′-GGGGCGGTCTCACATGCAAGCAAGAT-3′ and reverse 5′-CGGCTTGTCGACTCTTTAGTTTGGA-3′. The PCR product was purified by ethanol precipitation and digested with BsaI (Eco31I) and SalI. The expression vector pET21d_His was digested with NcoI-HF and SalI-HF. The digestion products were separated by electrophoresis in low-melting agarose gel. The band corresponding to linearized pET21_d His plasmid (5421 bp) was cut out, dephosphorylated, purified, mixed with the insert at the molar vector to insert ratio of 1:6, ligated with T4 DNA ligase, and transformed into competent *E. coli* TOP10. The recombinant pET21dHis_TP-84_28 clones (Appendix A) were screened by colony PCR using the primers 5′-GATGCGTCCGGCGTAGA-3′ and 5′-TGCTAGTTATTGCTCAGCGG-3′, SalI/StuI restriction analysis, and DNA sequencing. DNAs from “positive” clones were used for plasmid DNA purification and transformed into three types of competent cells: *E. coli* BL21(DE3), *E. coli* ArcticExpress(DE3), and *E. coli* BL21(DE3)pLysS. Clones were grown in LB medium supplemented with ampicillin (100 μg/mL) to an OD of 0.8 at 600 nm, and induced by the addition of 1 mM IPTG. Samples were collected after 1, 2, 3, 4, and 24 h after induction and analyzed using SDS-PAGE gels. The most efficient clone was selected for expression in 1 L of the LB medium.

#### 4.2.2. Recombinant TP84_28 Endolysin Purification

After 4 h of cultivation, the cells (1 g) were pelleted by centrifugation (5000× *g*, 4 °C, 15 min) and resuspended in 10 mL of buffer 1 (Tris-HCl pH 7.0, 500 mM NaCl, 0.5 mg/mL lysozyme) and disrupted using an ultrasonic homogenizer. The TP84_28 endolysin was purified from the soluble cytoplasm fraction by polyethyleneimine (PEI) precipitation of contaminating nucleic acids and acidic proteins (final concentration: 0.3%) by stirring at 4 °C for 16 h. The mixture was then centrifuged and the supernatant heated at 55 °C for 20 min. The precipitated *E. coli* proteins were removed by centrifugation (12,000× *g* for 30 min). The supernatant was extensively dialyzed to the buffer 1 for 16 h at 4 °C using 15 kDa cut-off membranes, and then filtered and subjected to Ni–NTA Sepharose next-generation chromatography (NGC) according to the manufacturer’s instructions. The recombinant TP84_28 endolysin was eluted in the range of 300–400 mM imidazole in the buffer 1. Finally, recombinant TP84_28 endolysin was purified on Superdex 16/60 75 pg using NGC and concentrated by an Amicon Ultra-4 filter (Millipore, Burlington, MA, USA) with a cut-off threshold of 10 kDa. The preparation was stored at 4 °C in the short term and at −20 °C for long-term storage. It was dialyzed against storage buffer S (50 mM Tris-HCl, pH 7.5, 150 mM NaCl, and 50% glycerol). 

### 4.3. Native TP84_28 Endolysin Production

#### 4.3.1. Bacterial Culture and Bacteriophage TP84-Mediated Cell Lysis

The bacterial host strain *G. stearothermophilus* strain 10 was grown in TYM medium at 58 °C to an OD of approx. 0.6; fructose was added to 0.5%, as originally recommended for enhancing bacteriophage yield [3]. After 30 min, the bacteriophage TP-84 was added (MOI = 0.1) and further cultivated, until lysis of the host was achieved. Then, chloroform was finally added to a concentration of 0.3%. In order to complete lysis, the culture was incubated from 1 h to overnight at 4 °C. Then, bacterial debris was removed by centrifugation and the supernatant was preserved from protease action with the addition of 1 mM phenylmethylsulfonyl fluoride (PMSF).

#### 4.3.2. Protein Purification

The first column, containing 4 mL CM–Sepharose, equilibrated with 10 column volumes (CV) of buffer A (50 mM K/PO_4_ (pH = 6.5) and 1 mM EDTA) was used for adsorption of the TP84_28 endolysin directly from the clarified host cell lysate after bacteriophage infection (1 L). After a 10 CV wash-step, elution was conducted with 3 CV of buffer A1 (50 mM K/PO_4_ (pH 6.5), 1 mM EDTA, and 500 mM NaCl) and 3 CV of buffer A2 (50 mM K/PO_4_ (pH 6.5), 1 mM EDTA, and 1000 mM NaCl). The TP84_28 endolysin was eluted at 500 mM NaCl and the protein was dialyzed against buffer A and loaded onto a 4 mL CM-Sephadex C-25 column, which was balanced with 10 CV of buffer A. The flowthrough from the previous step was also added to the column, followed by 10 CV of wash-step with buffer A and elution with 5 CV of buffer A1 and 5 CV of buffer A2 (50 mMK/PO_4_ (pH 6.5), 1 mM EDTA, and 1000 mM NaCl). The native TP-84_28 endolysin was collected at 500 mM NaCl. Next, the preparation was applied on 1 mL Q-Sepharose resin equilibrated with buffer A1 (50 mM K/PO_4_ (pH = 6.5), 1 mM EDTA, and 500 mM NaCl). The flowthrough was collected, since the column served as a “negative” step, under conditions where TP84_28 endolysin did not bind. The active TP84_28 endolysin was then filtrated through a VivaSpin Turbo Membrane at 100 MWCO in order to eliminate phage particles by centrifugation at 4000× *g* for 15 min. The filtrate was collected and concentrated at 10 MWCO at 4000× *g* for 15 min and dialyzed against the storage buffer S.

### 4.4. Western Blotting Analysis

Purified recombinant TP84_28 endolysin was separated by SDS-PAGE and electroblotted onto a PVDF membrane [32]. The membrane was probed with murine monoclonal anti-polyHistidine antibodies and conjugated with horseradish peroxidase. The detected protein was visualized by adding a solution of peroxidase substrate. The procedure was described in detail previously [33].

### 4.5. LC–MS–MS/MS Analysis

A liquid chromatography coupled to tandem mass spectrometry (LC–MS–MS/MS) assay was performed at the mass spectrometry laboratory (IBB PAS, Warsaw, Poland) [33]. The method consisted of protein fragmentation through trypsin digestion, followed by liquid chromatography and measurement of the mass of the resultant peptides by mass spectrometry. The sum of peptide masses was compared with the expected mass of the entire protein. The expected mass of the protein was calculated based on the amino acid (aa) sequence of a protein provided in the available databases. The raw data were processed using Mascot Distiller followed by Mascot Search (Matrix Science, UK) against the predicted, derived reference peptide masses.

### 4.6. Analysis of Protein Activity

#### 4.6.1. Diffusion Test

The activity of the purified TP84_28 endolysin was assayed using the diffusion method. The purified protein (from 2 µg) was twofold serially diluted in a sterile buffer (20 mM K/PO_4_ (pH 7.4) and 500 mM NaCl) and spotted onto a solidified TYM agar plate (0.7% agar), overlaid on a TYM agar bottom plate (1.5% agar). The soft agar contained 100 μL of the host bacteria, *G. stearothermophilus* strain 10, from an overnight culture. The plates spotted with 10 µL samples were air dried and incubated overnight at 55 °C. The semi-quantitative scoring of the effect of the TP84_28 endolysin (cleared spots) was conducted 16 h after plating the cells.

#### 4.6.2. Zymogram Analysis

The enzymatic activity of the protein was determined through zymographic assay [34]. Purified TP84_28 endolysin was separated by electrophoresis using 12% SDS-PAGE gel that included the *G. stearothermophilus* strain 10 host cells. For each zymogram, a 300 mL culture of host cells was stopped at an OD of 0.6 and the resulting cell pellet was suspended in 300 µL of 50 mM Na/PO4 (pH 7.0) and added to an unpolymerized SDS-PAGE gel. After running the electrophoresis, the zymogram was washed in excess distilled water and incubated at room temperature for 3 h to allow renaturation of the proteins. Transparent bands appeared where the enzymatically active TP84_28 endolysin band was located. After the zymogram was developed, it was stained with Coomassie Brilliant Blue or methylene blue for further visualization [35].

#### 4.6.3. Turbidity-Reduction Assay 

A standardized turbidity-reduction assay (TRA) was modified from Donovan et al. [26] with the *G. stearothermophilus* host strain used as a substrate. Bacteria in the mid-exponential phase (OD = 0.6) and stationary phase (OD = 1.2) were harvested by centrifugation (3900× *g*, 10 min, 4 °C) and stored on ice for up to 4 h. Immediately prior to the assay, the cell pellets were suspended in buffer R (50 mM Na/PO_4_ (pH 7.4) and 150 mM NaCl) to an OD of approximately 0.7. Upon adding 10 μL of the enzyme solution to 90 μL of the bacterial cell suspension, the OD was measured spectrophotometrically over time using a plate reader equipped with a Tecan spectrophotometer. The reactions were stopped with the addition of 0.15 μg proteinase K; serial dilutions of the cell suspensions were plated on LB agar in duplicates. Colonies were counted after 24 h of incubation at 55 °C. The antibacterial activity was quantified as the relative inactivation in log units (log10(N_0_/Ni)) with N0 as the initial number of untreated cells and Ni as the number of residual cells counted after treatment.

### 4.7. Antibacterial Activity

Bacterial cell cultures (100 mL each) of *E. coli*, *G. stearothermophilus* strain 10, *B. stearothermophilus*, *Geobacillus* ICI, *B. subtilis*, *B. cereus*, *Thermus aquaticus* YT, and *Thermus thermophilus* HB27 were grown to the mid-exponential phase (OD = 0.7), harvested by centrifugation, washed with 100 mL of buffer R, and finally resuspended in 1 mL of buffer R to a final density of ±10^7^ CFU/mL. A standardized TRA with the above-listed bacteria as substrates was used. A mixture of 10 μL of TP84_28 endolysin solution and 90 μL of each bacterial cell suspension was further incubated at 37 or 55 °C, and shaken for 30 min. In the control reaction, only the buffer R was added. The OD was measured and changes in OD during the incubation were plotted over time using a recording plate reader with a Tecan spectrophotometer.

### 4.8. CD Spectroscopy—Thermostability

To measure the melting temperature (Tm), the ellipticity of recombinant TP84_28 endolysin was recorded at 200–260 nm in a J-815 circular dichroism (CD) spectrometer (Jasco Corporation, Tokyo, Japan). The protein melting temperatures were determined with a heating rate of 1 °C/min, an incubation time of 3 s, and a volume of 500 μL in a 1 mm light path quartz cuvette (Hellma, Jena, Germany). All measurements were performed in buffer R. The midpoint of the unfolding transition was determined by fitting to a sigmoid unfolding model using JASCO analysis software (Spectra Manager Ver. 1.54M). The assay was conducted at protein concentrations of 0.2 mg/mL (4.44 μM).

### 4.9. DSC Analysis

DSC measurements were carried out using MCS-DSC or VP-DSC microcalorimeters (Microcal, Inc., Northampton, MA, USA) in buffer R. The recombinant TP84_28 endolysin samples (0.2 mg/mL (4.44 μM) and 0.4 mg/mL (8.89 μM)) and buffer were degassed under vacuum before being loaded into the calorimeter, and the scans were run under an additional constant pressure of 2 × 10^5^ Pa. The samples and reference cells were initially calibrated with equal volumes of buffer R using three consecutive heating/cooling cycles from 30 to 105 °C and from 105 to 30 °C at 1 °C/min. Data deconvolution by means of baseline subtraction and curve fitting was performed by Origin_DCS software.

### 4.10. pH Effect Evaluation Using TRA

TRA was conducted as in Section 4.6.3, except that different pH values between 4 and 10 were used. The same buffer R composition was used while varying the pH in order to exclude any possible effect of different buffer compounds. Upon the addition of 10 μL of recombinant TP84_28 endolysin solution (4.44 μM) to 90 μL of the bacterial cell suspension (a final protein concentration of 0.44 μM), the OD was measured spectrophotometrically over time using a plate reader equipped with a Tecan spectrophotometer. The reactions were subjected to constant mixing in a plate shaker to avoid cell sedimentation.

### 4.11. Biofilm Survey 

The biofilms (control sample) were examined using a quantitative spectrophotometric microtiter plate assay. Overnight cultures of host *G. stearothermophilus strain 10*, *E. cloacae*, *A. baumannii*, *S. aureus*, *Salmonella sp.*, *P. aeruginosa*, and *S. pyogenes* were diluted 1:150 in TSBg. Next, 4 mL aliquots were added to 6-well flat-bottom polystyrene microplates and incubated for 24 h at 37 °C (for host 55 °C). The plates were washed once with 1 × PBS (4 mL) and the biofilm biomass was stained for 15 min with 0.1% crystal violet (Sigma-Aldrich) and washed with 1 × PBS twice (2 × 4 mL). For quantification, crystal violet was solubilized in 4 mL of 33% (vol/vol) acetic acid and the optical density at 600 nm (OD_600_) was determined using a Victor 3 multimode microplate reader. The samples were examined in duplicate; TSBg alone was used as the negative control. The OD_600_ values of TSBg were subtracted from the average OD_600_ of each strain [36]. 

### 4.12. Biofilm Disruption

The biofilms (treatment sample) were examined using a quantitative spectrophotometric microtiter plate assay. Overnight cultures of host *G. stearothermophilus strain 10*, *E. cloacae*, *A. baumannii*, *S. aureus*, *Salmonella* sp., *P. aeruginosa*, and *S. pyogenes* were diluted 1:150 in TSBg. Next, 4 mL aliquots were added to 6-well, flat-bottom polystyrene microplates (Nest Scientific Biotechnology), the recombinant TP_28 endolysin was added (2 µg to each well), and the samples were incubated for 24 h at 37 °C (for host 55 °C). Afterwards, the 6-well plate was incubated for 1 h at 55 °C (only for mezophilic bacteria). The plates were washed once with 1 × PBS (4 mL) and the biofilm biomass was stained for 15 min with 0.1% crystal violet (Harleco; EMD Millipore Chemicals) and washed twice with 1 × PBS twice (2 × 4 mL). For quantification, crystal violet was solubilized in 4 mL of 33% (vol/vol) acetic acid; the OD_600_ value was determined using a Spectra Wallac 1420 Victor 3 multimode microplate reader. The samples were examined in duplicate and TSBg alone was used as the negative control. The OD_600_ values of TSBg alone were subtracted from the average OD_600_ of each strain [36].

## 5. Conclusions

In this paper, we managed to obtain two lytic enzymes (native and recombinant) with lysing and growth-inhibiting activity against *G. stearothermophilus*. Both enzymes showed high thermal stability, even up to 100 °C. In addition, the recombinant enzyme TP84_28 was tested for antimicrobial activity against both Gram-positive and Gram-negative mesophilic and thermophilic bacteria. Significant lytic activity was observed on thermophilic and mesophilic Gram-positive bacteria and, to a lesser extent, on thermophilic and mesophilic Gram-negative bacteria. This may be due to the structure of the conserved domains as well as the spatial structure resembling chitin-recognition domains. Thermostable endolysin TP84_28 seems to be a promising, mild agent for disinfecting surfaces exposed to high temperatures, disrupting bacterial biofilms in the food industry, and non-antibiotic treatment in medicine.

## Figures and Tables

**Figure 1 ijms-23-07612-f001:**
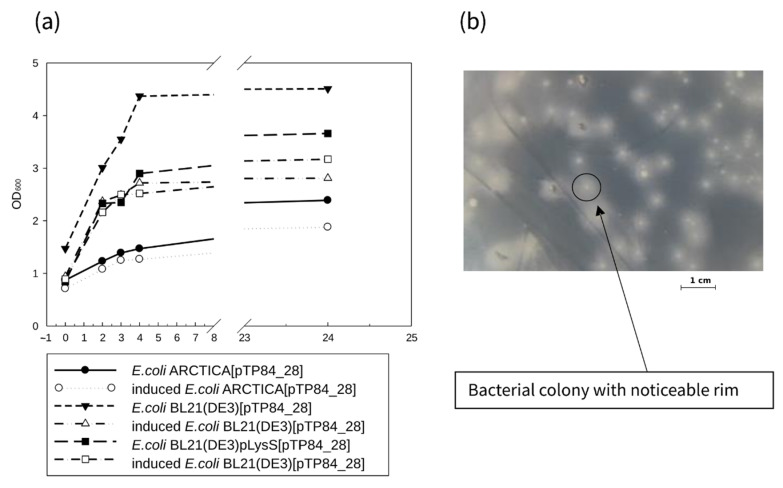
Expression of recombinant TP84_28 gene in three different *E. coli* cells. (**a**) Growth curves of bacterial clones, carrying recombinant pTP84_28 plasmids containing the TP84_28 gene in various *E. coli* expression strains: BL21(DE3), BL21(DE3) (pLysS), and ARCTICA. (**b**) *E. coli* BL21(DE3) (pTP84_28), grown on solid LB medium. Only this strain exhibited the characteristic rim around the bacterial colony, indicating interference of the cloned TP84_28 gene with the bacteria’s growth rate.

**Figure 2 ijms-23-07612-f002:**
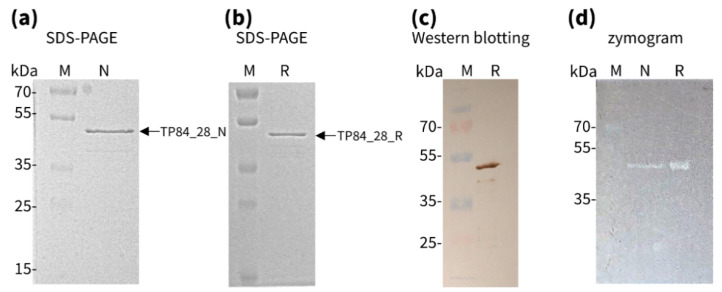
Identification of TP84-28 endolysin. (**a**) SDS-PAGE analysis of native TP84_28 proteins: M—protein ladder Prestained Plus; N—native TP84_28 protein; (**b**) SDS-PAGE analysis of recombinant TP84_28 protein: M—protein ladder Prestained Plus; R—recombinant TP84_28 protein; (**c**) Western blotting analysis of recombinant TP84_28 protein analysis using anti-His_6__tag antibodies: M—protein ladder Prestained Plus; R—recombinant TP84_28 protein; (**d**) Zymogram analysis of native and recombinant purified TP84_28 proteins: M—protein ladder Prestained Plus; N—native TP84_28 protein; R—recombinant TP84_28 protein.

**Figure 3 ijms-23-07612-f003:**
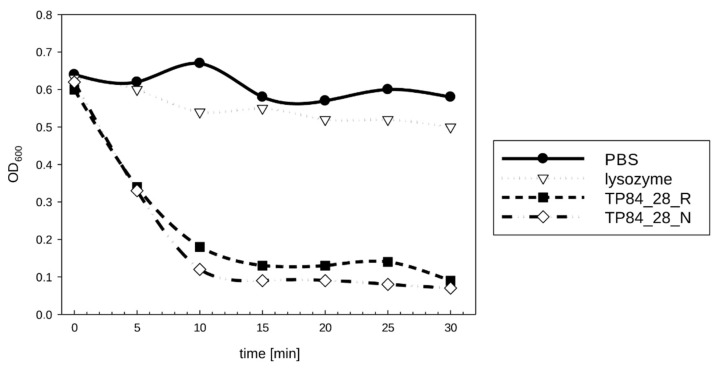
Comparison of the enzymatic activity of native and recombinant TP84_28 endolysin on *G. stearothermophilus* strain 10. Chicken egg lysozyme and PBS buffer were used as controls. The reaction was conducted at 55 °C for 30 min, an optimal temperature for TP84_28 endolysin. Thus, chicken egg lysozyme exhibits only traces of activity. TP84_28_N—native TP84_28 endolysin; TP84_28_R—recombinant TP84_28 endolysin.

**Figure 4 ijms-23-07612-f004:**
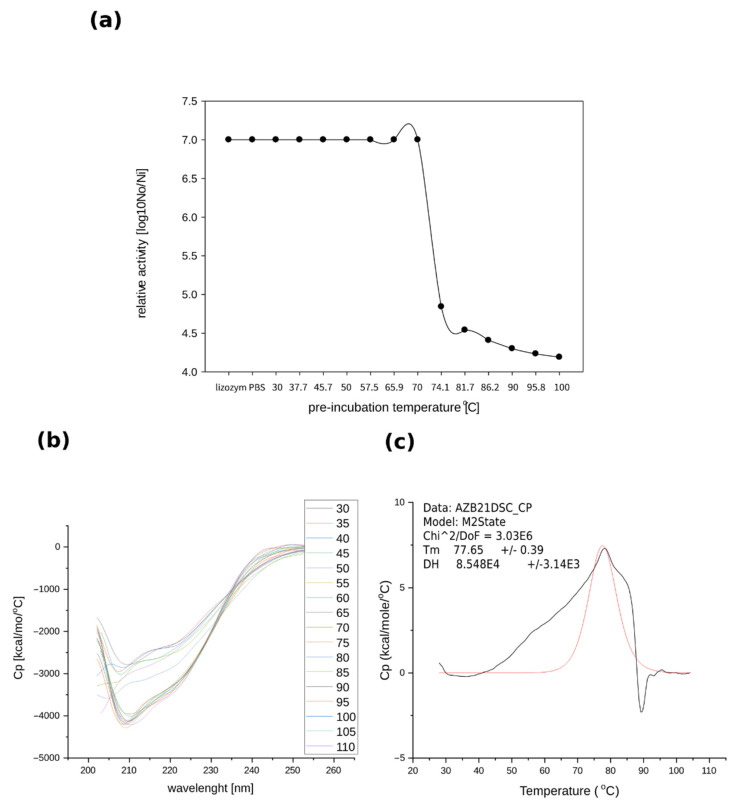
Determination of the thermal stability of recombinant TP84_28 endolysin. (**a**) TRA conducted on the pre-incubated enzyme and quantitative spectrophotometric measurements taken using a *G. stearothermophilus* strain 10 cell suspension; the dots represent the activity level of TP84_28 relative to the enzyme pre-incubation temperature (**b**) CD analysis; (**c**) DSC analysis; the heat-capacity curve of TP84_28 (black line) and the fit to the non-two-state model (red line).

**Figure 5 ijms-23-07612-f005:**
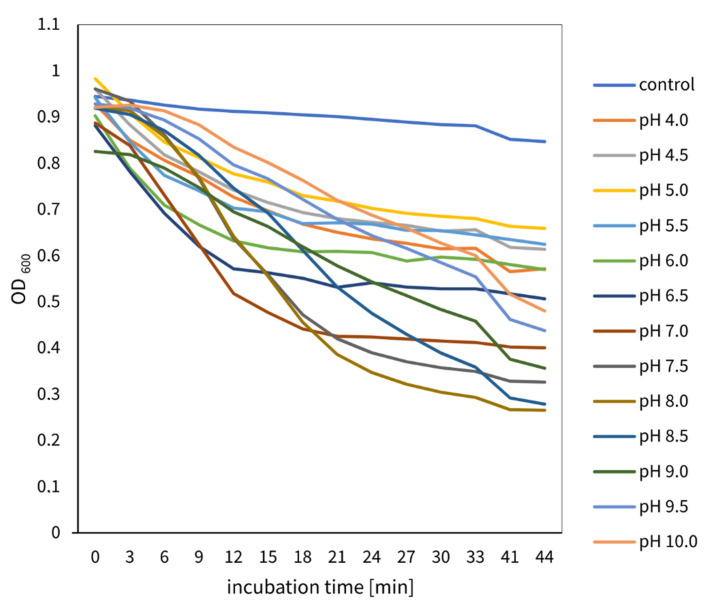
Effect of reaction pH on the muralytic activity of recombinant TP84_28 endolysin. The TRA assay, conducted as in Figure 4, tested reaction buffers with various pH values ranging from 4 to 10.

**Figure 6 ijms-23-07612-f006:**
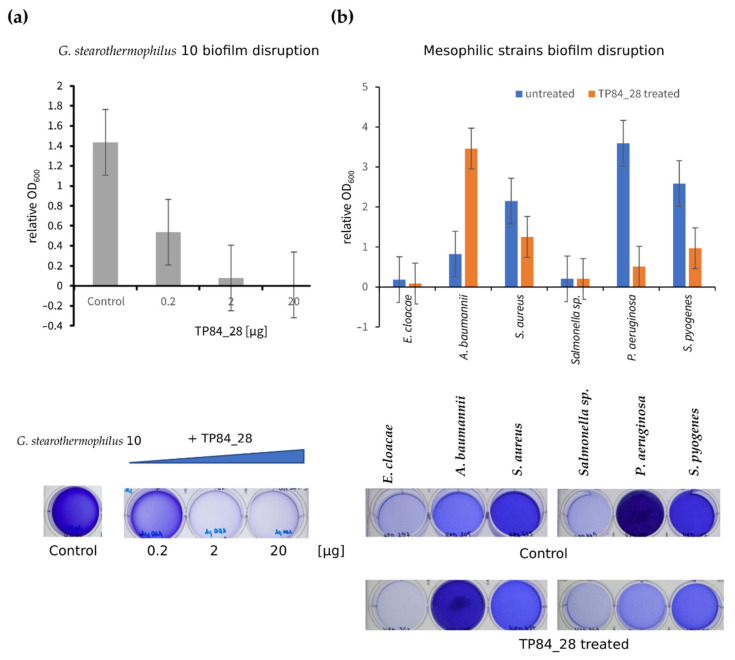
Disruption of bacterial biofilms. (**a**) *G. stearothermophilus* strain 10 biofilm disruption. The graph shows the relative OD_600_ nm value for the process control (PBS treated) with 24 h incubation at 55 °C with the addition of 0.2, 2.0, and 20 µg TP84_28 protein, respectively. Below the chart is a photo of biofilm staining with 0.1% crystal violet of the control and TP84_28 endolysin-treated samples. (**b**) Biofilm disruption in *E. cloacae*, *A. baumannii*, *S. aureus*, *Salmonella* sp., *P. aeruginosa*, and *S. pyogenes*. The graph shows the relative OD_600_ nm value for the process control of bacteria untreated with TP84_28 compared with samples incubated with 2 µg TP84_28 at 37 °C for 24 h and at 55 °C for 1 h. Below the chart, a photo of biofilm staining with 0.1% crystal violet of the control (PBS treated) and TP28 endolysin-treated samples.

**Table 1 ijms-23-07612-t001:** Enzymes involved in bacterial cell destruction during the lytic cycle of the bacteriophage.

Barrier	Enzyme	Enzyme Action		References
Phospholipid membrane	Holins (small proteins accumulate in the cytoplasmic membrane without impairing the cell during the expression of late bacteriophage genes)	Permeabilization of the cell membrane (allows endolysin to transfer from the cytoplasm to the cell wall) and preliminary degradation of the peptidoglycan layer		[7,8,9,10,11,12,13]
Peptidoglycan wall	Endolysins (peptidoglycan hydrolases and depolymerases) divide into four groups and are expressed in the final stage of bacteriophage infection	Glycosidases	Hydrolyzation of the β-1,4-glycosidic linkage of peptidoglycan behind the N-acetylglucosamine (GlcNAc) terminal and hydrolyzation of the glycosidic bonds at the end of N-acetylmuramyl (MurNAc)	[12,14,15,16,17,18]
		Amido- hydrolases	Digestion of the amide bonds between MurNAc and the peptide residue L-alanine of bacterial peptidoglycan (commonly found in bacteriophage endolysins)	
		Endo- peptidases	Hydrolyzation of peptide bonds in the peptides’ core directly attached to N-acetylmuramine acid or between peptides forming cross bridges and connecting the peptides	
		Lytic trans- glicosylases	Digestion of the β-1,4-glycosidic bond between residues MurNAc acid and GlcNAc acid with a different mechanism than glycosidases	
Polysaccharide layer	Polysaccharide depolymerases (present in a bacteriophage tail or diffused to the medium)	Glycanases	Degradation of the bacterial macromolecular carbohydrates in the capsule or structural polysaccharides (these help the bacteriophage in every stage of the lytic cycle)	[19,20]
		Polysaccharide lyases		

**Table 2 ijms-23-07612-t002:** Relative lytic activity of recombinant TP-84_28 endolysin on various bacterial substrates.

Species	Strain	Growth Temperature	Gram Staining	Activity of TP84_28	
				Turbidity Assay	Spot Assay
*G. stearothermophilus*	strain 10	55 °C/T	P	++	++
*Bacillus stearothermophilus*		55 °C/T	P	+	++
*Geobacillus*	ICI	60 °C/T	P	+++	+++
*Bacillus cereus*		37 °C/M	P	+/−	+
*Bacillus subtilis*		37 °C/M	P	+	+
*Escherichia coli*	DH11S/ATCC	37 °C/M	N	+/−	+
*Thermus thermophilus*	HB27/ATCC	65 °C/T	N	+/−	+
*Thermus aquaticus*	YT/ATCC	65 °C/T	N	+/−	+

## Data Availability

Data confirming the reported results are placed on an external drive and on password-protected clouds.

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
