# Peer review of "Cloning and Characterization of a Thermostable Endolysin of Bacteriophage TP-84 as a Potential Disinfectant and Biofilm-Removing Biological Agent"

_ijms, 2022, doi:10.3390/ijms23147612_

Round 1

Reviewer 1 Report

Although the study is valuable, it has some shortcomings. Various situations should be considered that will increase the research value. The introduction and method section should be modified with a clear understanding for readers and should be rearranged to be more understandable.

Typos should be corrected. The article should be accepted after minor revision.

Reviewer 2 Report

The authors described and characterized a thermostable lysin from phage. The work is quite straightforward and descriptive on the biochemical characterization of the lysin, however, relevance on the biological background and the application potent of such lysin molecule can be further strengthened. The advantage of this lysin over other known lysins was also largely ignored.

How active is the lysin against biofilm? For the host range, more relevant strains should be tested.
